# Effect of β-Glucan Supplementation on Growth Performance and Intestinal Epithelium Functions in Weaned Pigs Challenged by Enterotoxigenic *Escherichia coli*

**DOI:** 10.3390/antibiotics11040519

**Published:** 2022-04-13

**Authors:** Yuankang Zhou, Yuheng Luo, Bing Yu, Ping Zheng, Jie Yu, Zhiqing Huang, Xiangbing Mao, Junqiu Luo, Hui Yan, Jun He

**Affiliations:** 1Animal Nutrition Research Institute, Sichuan Agricultural University, Chengdu 611130, China; 2020214056@stu.sicau.edu.cn (Y.Z.); luoluo212@126.com (Y.L.); ybingtian@163.com (B.Y.); zpind05@163.com (P.Z.); yujie@sicau.edu.cn (J.Y.); zqhuang@sicau.edu.cn (Z.H.); acatmxb2003@163.com (X.M.); 13910@sicau.edu.cn (J.L.); yan.hui@sicau.edu.cn (H.Y.); 2Key Laboratory of Animal Disease-Resistance Nutrition, Chengdu 625014, China

**Keywords:** β-glucan, intestinal epithelium, inflammation, immunity, weaned pigs

## Abstract

Background: To examine the effect of β-glucan (BGL) supplementation on growth performance and intestinal epithelium functions in weaned pigs upon Enterotoxigenic *Escherichia coli* (ETEC) challenge. Methods: Thirty-two weaned pigs (Duroc × Landrace × Yorkshire) were assigned into four groups. Pigs fed with a basal diet or basal diet containing 500 mg/kg BGL were orally infused with ETEC or culture medium. Results: Results showed BGL tended to increase the average daily gain (ADG) in ETEC-challenged pigs (0.05 < *p* < 0.1). Dietary BGL supplementation had no significant influence on nutrient digestibility (*p* > 0.05). However, BGL improved the serum concentrations of immunoglobulin (Ig) A and IgG, and was beneficial to relieve the increasement of the concentrations of inflammatory cytokines such as the TNF-α and IL-6 upon ETEC-challenge (*p* < 0.05). Interestingly, BGL significantly increased the duodenal, jejunal and ileal villus height, and increased the jejunal ratio of villus height to crypt depth (V/C) upon ETEC challenge (*p* < 0.05). BGL also increased the activities of mucosal, sucrase and maltase in the ETEC-challenged pigs (*p* < 0.05). Moreover, BGL elevated the abundance of *Lactobacillus* and the concentration of propanoic acid in colon in the ETEC-challenged pigs (*p* < 0.05). Importantly, BGL elevated the expression levels of zonula occludins-1 (ZO-1) and mucin-2 (MUC-2) in the small intestinal mucosa upon ETEC challenge (*p* < 0.05). BGL also upregulated the expressions of functional genes such as the claudin-1, cationic amino acid transporter-1 (CAT-1), LAT-1, L amino acid transporter-1 (LAT1), fatty acid transport proteins (FATP1), FATP4, and sodium/glucose cotransporter-1 (SGLT-1) in the duodenum, and the occludin-1 and CAT-1 in the jejunum upon ETEC challenge (*p* < 0.05). Conclusions: These results suggested that BGL can attenuate intestinal damage in weaned pigs upon ETEC challenge, which was connected with the suppressed secretion of inflammatory cytokines and enhanced serum immunoglobulins, as well as improved intestinal epithelium functions and microbiota.

## 1. Introduction

Weaning is a critical challenge for mammalian animals including pigs. Abrupt changes in diet form and removal of the passive maternal protection in post-weaning piglets increases their susceptibility to diarrhea, resulting in a certain degree of damage to the intestinal structure and mucosal barrier functions [1,2]. ETEC is one of the most critical bacterial causes of intestinal diarrhea [3]. Upon adhering to the intestinal epithelium with their fimbriae, ETEC produces various enterotoxins that can take effect on the small intestinal epithelium and cause the secretion of fluids and electrolytes, which can lead to diarrhea [4]. Over the previous few decades, antibiotics have been extensively used to relieve diarrhea resulting from various bacteria and weaning stress in pig production [5,6]. However, drug residues in the product and the bacterial resistance against classical antibiotics are growing problems all over the world, and it is urgent to develop alternatives to traditionally used antibiotics [7,8].

The most extensive studied alternatives include probiotics, prebiotics, enzymes, acidifiers, and plant extracts [9]. Among the feed additives applied in pigs, prebiotics are regarded as preferable, since they can accelerate competitive exclusion of pathogenic microorganisms and selectively promote the colonization of beneficial microbes [10]. Among the known prebiotics, the mannan-oligosaccharide (MOS) and fructo-oligosaccharide (FOS) have been extensively tested in swine and poultry [11,12]. For instance, both in vivo and in vitro studies showed that in the presence of MOS and FOS, the enteric pathogens do not adhere to the epithelium but to the sugar compounds in the enterocoel, which obviously deceased their colonization and intestinal inflammation [13,14,15].

β-glucan (BGL) has also been known as an important natural prebiotic that is abundant in lentinan, cereals, and the yeast cell wall [16]. β-(1,3)-glucan can form a triple helix structure, which can facilitate the interaction of molecules and receptors and induce a biological effect. Because of the deficiency of β-glucanase in animals, BGL can escape the enzymatic digestion from the upper gastrointestinal tract and enter the hindgut as a fermentation substrate for microorganisms. Previous studies indicated that BGL can selectively stimulate the growth of beneficial bacteria and help to maintain the intestinal health [17,18]. Moreover, the immunomodulatory effects of BGL have also been well documented [19,20]. For instance, BGL isolated from the yeasts considerably promoted the production of serum cytokines and IgA in hens [21]. It is well known that differences in molecular weight and branching degree of glucan affect the solubility of BGL, which may ultimately affect their immune-modulating properties [22]. As compared to those without branches, BGLs consist of a (1,3)-β-linked backbone with β-(1,6) showed a higher immunomodulatory activity [23]. BGL can also act as a pathogen associated molecular pattern (PAMP), and dectin-1 is the main recognition receptor that is highly expressed in macrophages and dendritic cells [24].

Although numerous studies have indicated a positive role of BGL in regulating the intestinal health and immune functions, data regarding the effect of BGL on the intestinal epithelium functions of the weaned pig exposure to ETEC are scarce. The aim of this study was to examine whether dietary BGL supplementation could relieve ETEC-induced intestinal inflammation and epithelium damage in weaned pigs. This study will also help in the understanding of the mechanisms behind the BGL-regulated intestinal health.

## 2. Results

### 2.1. Effect of BGL on Growth Performance and Nutrients Digestibility in Weaned Pigs upon ETEC Challenge

As shown in Table 1, no significant differences were observed on ADFI, ADG and F: G among all groups before the ETEC challenge (*p* > 0.05). However, ETEC challenge significant decreased the ADG in the weaned pigs (*p* < 0.05). BGL supplementation relieved the decrement of daily gain (ADG), but there was no significant difference in the ETEC-challenged pigs (*p* > 0.05). In addition, no significant differences were observed on the four treatments for nutrients digestibility (Table 2).

### 2.2. Effect of BGL on Serum Immunoglobulins and Inflammatory Cytokines in Weaned Pigs upon ETEC Challenge

As shown in Figure 1, BGL supplementation increased the concentration of IgA in the non-challenged as well as in the ETEC-challenged pigs (*p* < 0.05). BGL supplementation increased the concentration of IgG in the ETEC-challenged pigs (*p* < 0.05). ETEC enhanced the concentration of IL-1β and IL-6 in the serum (*p* < 0.05). However, BGL supplementation greatly relieved the improvement of the concentration of TNF-α and IL-6 caused by the ETEC challenge (*p* < 0.05).

### 2.3. Effect of BGL Supplementation on Intestinal Morphology and Mucosal Enzyme Activity in Weaned Pigs upon ETEC Challenge

As shown in Table 3 and Figure 2, ETEC challenge acutely reduced the jejunal villus height (*p* < 0.01). Nevertheless, pigs fed with BGL supplementation enormously improved the jejunal villus height and the ratio of V/C upon ETEC challenge (*p* < 0.05). Furthermore, BGL supplementation also enhanced duodenal and ileal villus height upon ETEC challenge (*p* < 0.05). ETEC challenge decreased the activities of maltase, lactase, and sucrase in the jejunum (Table 4). However, BGL supplementation significantly relieved the reduction of their activities upon ETEC challenge (*p* < 0.05). BGL supplementation also enhanced the duodenal and ileal activity of maltase upon ETEC challenge (*p* < 0.05).

### 2.4. Effect of BGL Supplementation on Expressions of Critical Genes Involved in Intestinal Epithelium Functions

As shown in Figure 3, the ileal expression levels of ZO-1 obviously decreased upon ETEC challenge (*p* < 0.05). Conversely, BGL supplementation greatly increased the duodenal, jejunal and ileal expression levels in the ETEC-challenged pigs (*p* < 0.05). Pigs fed with BGL improved the duodenal expression levels of claudin-1 and jejunal expression levels of occludin upon ETEC challenge (*p* < 0.05). Moreover, BGL elevated the expression levels of functional genes such as the MUC2, CAT1, LAT1, FATP1, and FATP4 in the duodenum, and increased the jejunal expression levels of MUC2 and CAT1 upon ETEC challenge (*p* < 0.05).

### 2.5. Effect of BGL Supplementation on Intestinal Microbial Populations in Weaned Pigs upon ETEC Challenge

ETEC challenge enriched the abundance of *Escherichia coli* in cecum (*p* < 0.05). BGL supplementation excellently enhanced the abundance of *Lactobacillus* upon ETEC challenge (*p* < 0.05). BGL supplementation also improved the concentration of propanoic acid in the ETEC-challenged pigs (Table 5).

## 3. Discussion

As a commercially feasible prebiotic, BGL can escape from enzymatic digestion in the upper intestine, but can be fermented by various microorganisms in the hindgut to produce various short-chain fatty acids that contribute to the maintenance of the intestinal health [25,26]. Importantly, the BGL has long been known as a stimulator of cellular immunity [27]. In this study, the ETEC challenge significantly reduced the ADG in the weaned pigs, but BGL had a tendency to increase the ADG in the ETEC-challenged pigs. The beneficial effects of dietary BGL supplementation on growth performance have also been observed in previous studies using different animal species [28,29,30].

Immunoglobulins, also known as antibodies, were a class of glycoproteins secreted by plasma cells. They play a key role in the immune response, and they can specifically recognize and bind to particular antigens, such as bacteria or viruses, and assist with their destruction. Currently, five classes (isotypes) of immunoglobulins (IgM, IgG, IgA, IgD, and IgE), were identified according to their heavy chain subunits [31]. Amongst the five classes, IgG is the primary immunoglobulin in serum, which can promote phagocytosis of mononuclear macrophages and neutralize the toxicity of bacterial toxins [32], whereas IgA is the second abundant immunoglobulin in serum [33]. In this study, BGL supplementation maintained a high level of serum IgA and IgG upon ETEC challenge, indicating an immune enhancement in the pigs upon ETEC challenge. As several common pro-inflammatory cytokines, TNF-α, IL-1β and IL-6, are widely considered as the biomarkers of inflammatory response or inflammation [34]. In the present study, the concentrations of them in the serum were elevated upon ETEC challenge, which is consistent with previous studies on pigs [12,15]. However, BGL supplementation greatly relieved the increase of the serum concentration of the TNF-α and IL-6 upon ETEC challenge.

The intestinal tract is the foremost functional site for monogastric animals to digest and absorb nutrients. The intestinal mucosa villus height and crypt depth are popular indexes to characterize intestinal morphology [35]. However, the intestinal morphology can be impaired upon various bacterial or viral infections [36,37]. In this study, the jejunal villus height was significantly reduced upon ETEC challenge, a finding that is in agreement with current studies on pigs [38]. However, BGL supplementation increased the jejunal villus height and the ratio of V/C upon ETEC challenge. Moreover, BGL treatment both elevated the jejunal mucosa liveness of lactase, sucrase, and maltase upon ETEC challenge. The beneficial effect of BGL on gut health may result from its sugar-chain structure, as a wide variety of prebiotics were shown to prevent enteric pathogens (e.g., ETEC) from attaching to the sugar compounds in the intestinal epithelium, which efficaciously cut down their colonization and intestinal inflammation [39,40,41]. Moreover, the BGL suppressed production of inflammatory factors (for example TNF-α and IL-6) and might also contribute to the improved integrity of the intestinal epithelium upon ETEC challenge, as they were previously reported to induce mucosa atrophy and apoptosis of the intestinal epithelial cells [42].

Tight junctions (TJ) are crucial to retaining the intestinal barrier, and disruption of the intestinal TJ may result in elevated permeability of the intestinal epithelium and entering of harmful substances into the blood [43,44]. In the present study, in pigs challenged with ETEC, the expression levels of the major TJ protein ZO-1 in duodenum and ileum were reduced; however, BGL supplementation significantly increased their expression levels. Moreover, BGL also improved claudin-1 and occludin expression levels in the duodenum and jejunum upon ETEC challenge, respectively. Claudin-1 and occludin are another two critical TJ proteins which are widely expressed in epithelial tissues. A previous study has shown that claudin-1- deficient mice died due to skin wrinkling, severe dehydration and an increase in epidermal permeability [45]. Occludin promotes the formation of reticular TJs and reduces the permeability of ions and macromolecules to enhance the intestinal barrier [46]. The elevation of their expression levels by BGL supplementation indicated an improved integrity of the intestinal barrier upon ETEC challenge. We also explored the expression levels of several key molecules that are associated with the intestinal barrier functions. MUC2 is a major intestinal secretory mucin and acts as a major component of the mucus layer. Importantly, the MUC2 was found to limit the intestinal luminal load by protecting against the accumulation of some less-adherent pathogens (A/E bacteria) [47]. Moreover, the generation of NO is extremely dependent on L-arginine, and the CAT1 is a major L-arginine transmembrane transporter in endothelial cells [48]. In this study, the jejunal expression levels of MUC2 and CAT1 was significantly improved due to BGL treatment after ETEC-challenge. In addition, the duodenal expression levels of LAT1 and FATP4 showed the same result. The LAT1 is one of the major bidirectional transporters of large neutral amino acids, and is responsible for the transportation of Leu into muscle cells [49], whereas the FATP4 is responsible for the uptake of long-chain fatty acids [50]. Elevated expressions of these critical functional genes indicated that BGL treatment is beneficial to enhance intestinal integrity and epithelial functions in weaned piglets under ETEC challenge.

The impact of the intestinal microbiota on health and disease is increasingly emerging. Like with other prebiotics, BGLs are resistant to enzymatic hydrolysis in the small intestine, and thence the majority enters the large intestine in an integral form [51]. In this experiment, BGL supplementation increased the abundance of *Lactobacillus* in the cecum upon ETEC challenge. The *Lactobacillus* is one of the most critical beneficial microorganisms in the gut, which is not only capable of promoting secretion of sIgA from the lamina propria to the surface of the intestinal epithelium, but also capable of activating the macrophages via the TLR2 signaling pathway [52,53]. Prebiotics can be used as substrate for microbial fermentation, causing production of various SCFAs (e.g., acetate, propionate, butyrate) [54]. A previous study has indicated that SCFAs can suppress the ETEC infection in the intestine by reducing virulence gene expression, flagella movement, and colonization [55]. Moreover, the concentration of propanoic acid was elevated by BGL upon ETEC challenge. The increased concentration of propanoic acid was found to increase blood flow and thereby promote the proliferation of intestinal epithelial cells [56]. Moreover, propanoic acid can also enhance the intestinal barrier by increasing the TJs expression levels, such as claudin-1, claudin-8, and occludin [57]. The present experiment was approved a potential mechanism by which the BGL-improved intestinal barrier functions.

## 4. Materials and Methods

The animal experiment was approved by the Institutional Animal Care and Use Committee of Sichuan Agricultural University (No. 20181105). Glucan product (β-glucan ≥ 60%) was kindly provided by SYNLGHT BIO Co., Ltd. (Guangzhou, China) Pathogenic *Escherichia coli* (ETEC, O149:K91, K88ac) was purchased from the China veterinary culture collection center with a CVCC no. 225 (Beijing, China).

### 4.1. Animal Diets and Experimental Design

A total of 32 commercial piglets (DLY) weaned at 21 d (with an average body weight of 6.82 ± 0.16 kg) were blocked by weight and allocated as a 2 (BGL) × 2 (ETEC) factorial design to four treatments (*n* = 8) composed of CON (the basal diet), BGL (500 mg/kg BGL) and ECON (the basal diet and challenged by ETEC), EBGL (500 mg/kg BGL and challenged by ETEC). The basal diet (Table 6) was designed to satisfy the swine nutrient requirements recommended byNRC2012 [58]. Pigs were individually housed in 1.5 × 0.7 m^2^ metabolism cages and had free access to feed and water. The temperature was controlled between 27–30 °C and the relative humidity was kept at 65 ± 5%. The experiment lasted for 28 d. The ETEC-challenged pigs were orally infused with 100 mL Luria-Bertani (LB) medium (containing 1 × 10^10^ CFU/mL ETEC) on day 26, and other pigs were infused with an equal volume of LB medium.

### 4.2. Growth Performance Evaluation

Piglets were weighed on day 1, 26, and 29 after 12 h fasting. The average daily gain (ADG), average daily feed intake (ADFI), and feed efficiency (F/G) were calculated by recording the feed intake and body weight gain during the experimental period.

### 4.3. Sample Collection

Uncontaminated fecal samples were collected straightway after defecation on days 22 to 25, and mixed with 10% H_2_SO_4_ (10 mL H_2_SO_4_ per 100 g of fresh fecal) and 1–2 drops of toluene. The four feeds and uncontaminated fecal samples were dried at 65 °C until constant weight, and then stored in powder form of a size that it could pass through a 1-mesh screen. On day 29, blood samples were collected by precaval vein puncture, and serum samples were centrifuged at 3500× *g* for 15 min after standing for 30 min at 4 °C, fecal and serum samples were stored at –20 °C until the next analysis. After blood collection, the pigs were anesthetized by intravenous injection with sodium pentobarbital (200 mg/kg BW), and segments of the duodenum, jejunum and ileum (about 2–4 cm in the middle sections) were isolated and rinsed softly with phosphate buffered saline (PBS), and then fixed in 4% paraformaldehyde solution for intestinal morphological analysis. Moreover, cecal digesta samples were collected into sterile tubes, and the duodenal, jejunal, and ileal mucosa samples were scraped with a glass slide and quick-freeze using liquid N_2_, followed by the preservation at −80 °C until further analysis.

### 4.4. Apparent Total Tract Nutrient Digestibility Analysis

Diet and fecal samples were used for the nutrient digestibility analysis, and the acid insoluble ash (AIA) was regarded as an endogenous indicator. The dry matter (DM), crude protein (CP), ether extract (EE), and Ash contents were measured according to AOAC methods [59]. Moreover, the gross energy (GE) was measured by an adiabatic bomb calorimeter (LECO, St. Joseph, MI, USA). All apparent digestibility of nutrients were calculated by the following formula:(1)$$\mathrm{Apparent}\;\mathrm{digestibility}\;\mathrm{of}\;a\;\mathrm{nutrient}\;\left(\%\right)=100-100\times \frac{\mathrm{digesta}\;\mathrm{nutrient}\;\times \;\mathrm{diet}\;\mathrm{AIA}}{\;\mathrm{diet}\;\mathrm{nutrient}\;\times \;\mathrm{digesta}\;\mathrm{AIA}}$$

### 4.5. Serum Proinflammatory Cytokines and Immunoglobulin Detection

The concentration of proinflammatory cytokines (TNF-α, IL-1β and IL-6) and immunoglobulin (IgG, IgM, and IgA) in serum were determined by Enzyme Linked Immunosorbent Assay (ELISA) kits (Shanghai Meimian Biotechnology Co., Ltd., Shanghai, China). There was less than 10% variation of intra-assay and 12% variation of inter-assay coefficients for each assay.

### 4.6. Histomorphology Analysis of Each Intestinal Segment

The intestinal segment fixed with 4% paraformaldehyde was dewaxed by graded anhydrous ethanol, then stained with hematoxylin and eosin (H&E), dehydrated by graded anhydrous ethanol again, and then sealed with neutral resin. Finally, the crypt depth and villus height of the samples were calculated using an image processing and analysis system (Image-Pro Plus 6.0), and their V/C was calculated. The procedure and statistical method of histomorphology analysis were followed by Wan’s. A total of 10 intact, well-oriented villus heights and corresponding crypt depths were obtained per section, and each intestinal segment was analyzed in triplicate [60].

### 4.7. Enzyme Activity

Using the frozen saline as the homogenization medium, the duodenal, jejunal and ileal mucosa were made into 10% homogenate, and then the supernatants were collected after centrifugation at 4000× *g* for 20 min. The lactase, maltase and sucrase were determined by the method of enzyme linked immunosorbent assay (ELISA), and the kits purchased from Shanghai Enzyme-linked Bio-technology Co., Ltd. (Shanghai, China), and kits as follows: lactase (Porcine Lactase ELISA Kit ml712060), sucrase (Porcine Sucrase ELISA Kit ml712026), and maltase (Porcine Maltase ELISA Kit ml712030). All procedures were performed according to the instructions of the kits.

### 4.8. Caecal Microbiological Analysis

About 0.2 g cecum digesta was processed using the Stool DNA Kits (Omega Bio-Tek, Doraville, CA, USA) to total DNA extraction for quantification real-time PCR, which was executed by conventional PCR on the CFX96 Real-Time PCR Detection system (Bio-Rad Laboratories, Hercules, CA, USA). Total bacteria were assessed by the reaction which runs in a total volume of 25 μL with 12.5 μL SYBR Premix Ex Taq (2× concentrated), 1 μL of forward and reverse primers respective (100 nM), 2 μL DNA, and 8.5 μL of RNase-Free ddH_2_O. This application program entailed 95 °C for 25 s; followed by 40 cycles of 95 °C for 5 s and 64.5 °C for 25 s; and then a final melting-curve for SYBR Green tests. *Lactobacillus*, *E. coli*, *Bacillus and Bifidobacterium* were tested by the SuperReal PreMix (Probe) kit (Tiangen Biotech Co., Ltd., Beijing, China). The reaction run in a total volume of 20 μL with 10 μL 2 × Super Real PreMix (Probe), 0.6 μL of forward and reverse primers (100 nM) respective, 0.4 μL probe (100 nM), 2 μL DNA and 6.4 μL of RNase-Free ddH_2_O. All reaction was included in one cycle of pre-denaturation at 95 °C for 15 min; fifty cycles of denaturation at 95 °C for 3 s; annealing and extension at 53 °C for 25 s. The Cycle threshold (Ct) values and baseline settings were determined by automatic analysis settings, and the copy numbers of the target group for each reaction were calculated from the standard curves of plasmid DNA produced by a 10-fold serial dilution of (1 × 10^1^ to 1 × 10^9^ copies/μL).

### 4.9. Metabolite Concentrations in Cecal Contents

The concentrations of SCFA (acetic acid, propanoic acid, and butyric acid) were determined by a gas chromatograph (VARIAN CP-3800, Varian, Palo Alto, CA, USA; capillary column 30 m × 0.32 mm × 0.25 μm film thickness) according to the previous method [61]. After centrifuging (12,000× *g* for 10 min), the supernatant was mixed with 0.2 mL metaphosphoric acid and 23.3 μL 210 mmol/L crotonic acid in a new tube, and those mixtures were centrifuged in the same conditions again after 30 min incubation at 4 °C. 1 μL of the supernatant were analyzed through the gas chromatograph. The polyethylene glycol column was operated with highly purified N_2_ as carrier gas at 1.8 mL/min.

### 4.10. Isolation and Reverse Transcription of RNA from Intestinal Mucosa and q-PCR

Following the manufacturer’s instructions, total RNA was extracted from a bit of duodenal, jejunal and ileal mucosa samples. All mucosa samples were homogenized with 1 mL of RNAiso Plus (Takara Biotechnology Co., Ltd., Dalian, China), and the concentration and fineness of total RNA were measured by a spectrophotometer (NanoDrop 2000, Thermo Fisher Scientific, Inc., Waltham, MA, USA). Approximately 1 μg total RNA was reverse transcribed into cDNA according to the protocol of the PrimeScript™ RT reagent kit with gDNA Eraser (Takara Biotechnology Co., Ltd., Dalian, China). This process was as follows: (1) 37 °C for 15 min, (2): 85 °C for 5 s. The expression level of the target genes was quantified by q-PCR, and the oligonucleotide primer sequences were shown in Appendix A. qPCR was performed with the SYBR^®^ Green PCR I PCR reagents (Takara Bio Inc., Dalian, China) using the same PCR system mentioned above. The reaction mixture (total volume of 10 μL) composed of 5 μL SYBR Premix Ex Taq II (Tli RNaseH Plus), 0.4 μL forward primer and reverse primer, 1 μL cDNA and 3.2 μL RNase-Free ddH_2_O. The procedure of q-PCR is as follows: 95 °C for 30 s, followed by 40 cycles: at 95 °C for 5 s and 60 °C for 30 s. The mRNA relative expression level of target genes was standardized by the housekeeping gene β-actin, which was calculated based on the 2^–ΔΔCt^ method [62].

### 4.11. Statistical Analysis

The data was analyzed by two-way ANOVA with the general linear model (GLM) procedure of SPSS as a two (BGL) × 2 (ETEC) factorial design. *p*-value < 0.05 was considered as significant and the *p*-value variation from 0.05 to 0.1 was considered as a significant trend. Duncan’s multiple range test was used based on the analysis of ANOVA, which showed a significant difference. All data were analyzed by SPSS 27.0 (IBM, Chicago, IL, USA). Results are expressed as means with their standard errors.

## 5. Conclusions

In conclusion, our results indicated a positive effect of dietary BGL supplementation on the growth performance and intestinal health in the weaned pigs upon ETEC challenge. The mechanisms behind its action may be connected with the suppressed secretion of inflammatory cytokines, improved immunity and intestinal morphology, as well as changes of the microbial fermentation.

## Figures and Tables

**Figure 1 antibiotics-11-00519-f001:**
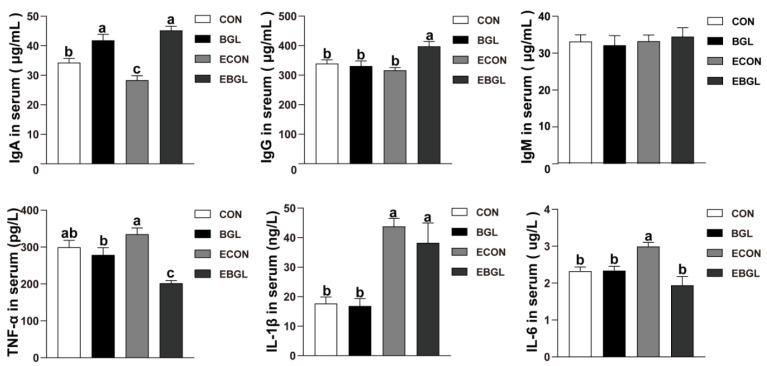
Effect of BGL on serum immunogloblins and inflammatory cytokines in weaned pigs upon ETEC challenge TNF-α, tumor necrosis factor-α; IL-1β, interleukin1-β; IL-6, interleukin-6; IgA, immunoglobulins A; IgG, immunoglobulins G; IgM, immunoglobulins M. a, b, c mean values within a row with unlike superscript letters were significantly different (*p* < 0.05). CON pigs were fed with a basal diet; BGL pigs were fed with a BGL containing diet (500 mg/kg); ECON pigs were fed with a basal diet and challenged by ETEC; EBGL pigs were fed with a BGL containing diet (500 mg/kg) and challenged by ETEC.

**Figure 2 antibiotics-11-00519-f002:**
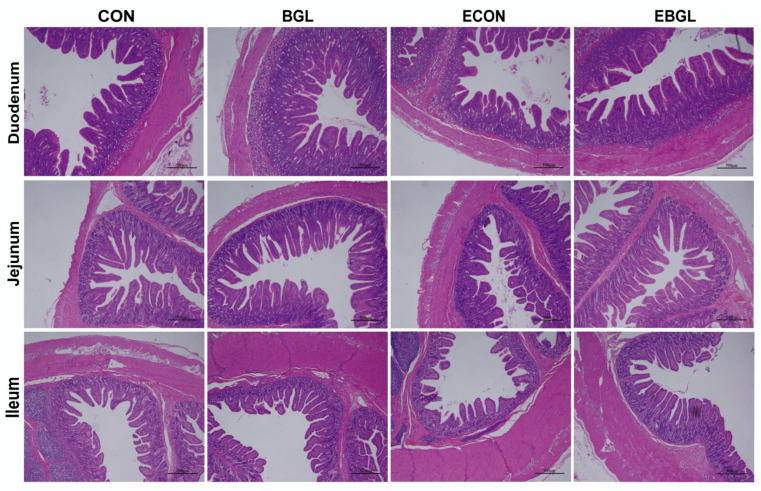
Effect of BGL supplementation on intestinal morphology in weaned pigs upon ETEC challenge (H&E; × 40) CON pigs were fed with a basal diet; BGL pigs were fed with a BGL containing diet, 500 mg/kg; ECON pigs were fed with a basal diet and challenged by ETEC; EBGL pigs were fed with a BGL containing diet (500 mg/kg) and challenged by ETEC. Duodenum, jejunum and ileum in the CON group revealed a normal appearance with regular intestinal villus structure; duodenum, jejunum and ileum in the BGL group, no obvious damages were found. However, in the ECON group, the intestinal lesions were obvious and some intestinal villi were necrotic and shed, or even disappeared, especially in the jejunum; in the duodenum, jejunum and ileum in the EBGL group there were no obvious intestinal lesions and only a few of the intestinal villi were necrotic and shed.

**Figure 3 antibiotics-11-00519-f003:**
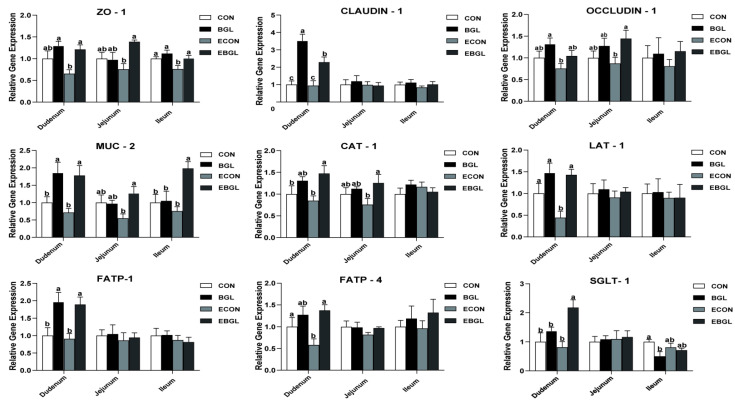
Effect of BGL supplementation on mucosal gene expressions in weaned pigs upon ETEC challenge ZO-1, zonula occludens-1. SGLT-1, sodium/glucose cotransporter-1; CAT-1, cationic amino acid transporter-1; LAT-1, L amino acid transporter-1; FATP, fatty acid transport proteins; a, b, c mean values within a row with unlike superscript letters were significantly different (*p* < 0.05). CON pigs were fed with a basal diet; BGL pigs were fed with a BGL containing diet, 500 mg/kg; ECON pigs were fed with a basal diet and challenged by ETEC; EBGL pigs were fed with a BGL containing diet (500 mg/kg) and challenged by ETEC.

**Table 1 antibiotics-11-00519-t001:** Effect of BGL supplementation on ADFI and ADG in weaned pigs upon ETEC challenge.

ITEM	Treatments				SEM	*p*-Value		
	CON	BGL	ECON	EBGL		BGL	ETEC	Interaction
1–25 d								
ADFI (g/d)	427.11	429.16	459.58	459.77	22.47			
ADG (g/d)	280.47	267.2	286.48	302.13	15.61			
F: G	1.58	1.62	1.56	1.55	0.04			
25–28 d								
ADFI (g)	631.68	602.26	537.85	597.22	24.96	0.78	0.35	0.4
ADG (g)	519.9 ^a^	476.67 ^ab^	361.33 ^b^	511.11 ^ab^	27.9	0.31	0.24	0.08
F: G	1.25	1.3	1.53	1.24	0.07	0.39	0.41	0.22

ADFI, average daily feed intake; ADG, average daily gain; F: G, feed: gain ratio. Mean and total SEM are listed in separate columns (*n* = 8). a, b mean values within a row with unlike superscript letters were significantly different (*p* < 0.05). CON pigs were fed with a basal diet; BGL, pigs were fed with a BGL containing diet (500 mg/kg); ECON pigs were fed with a basal diet and challenged by ETEC; EBGL pigs were fed with a BGL containing diet (500 mg/kg) and challenged by ETEC.

**Table 2 antibiotics-11-00519-t002:** Effect of BGL supplementation on nutrients digestibility in weaned pigs.

ITEM	Treatments				SEM	*p*-Value
	CON	BGL	ECON	EBGL		
DM (%)	88.43	88.69	88.98	89.17	0.40	0.93
CP (%)	86.53	86.49	87.18	87.62	0.68	0.93
EE (%)	84.96	84.89	86.34	86.56	0.58	0.65
Ash (%)	67.65	70.6	69.55	70.91	0.86	0.58
GE (%)	88.56	88.76	89.17	89.27	0.43	0.94

DM, dry matter; CP, crude protein; EE, ether extract; GE, gross energy. Mean and total SEM are list in separate columns (*n* = 8). CON pigs were fed with a basal diet; BGL pigs were fed with a BGL containing diet (500 mg/kg); ECON pigs were fed with a basal diet and challenged by ETEC; EBGL pigs were fed with a BGL containing diet (500 mg/kg) and challenged by ETEC.

**Table 3 antibiotics-11-00519-t003:** Effect of BGL supplementation on intestinal morphology in weaned pigs upon ETEC challenge.

ITEM	Treatments				SEM	*p*-Value		
	CON	BGL	ECON	EBGL		BGL	ETEC	Interaction
Duodenum								
Villus height, μm	402.98 ^ab^	461.37 ^a^	378.85 ^b^	454.45 ^a^	13.79	0.02	0.55	0.74
Crypt depth, μm	135.42	139.42	156.99	154.57	5.4	0.95	0.1	0.78
V:C	3.12 ^ab^	3.52 ^a^	2.73 ^b^	3.19 ^ab^	0.12	0.07	0.12	0.92
Jejunum								
Villus height, μm	413.89 ^a^	442.74 ^a^	299.70 ^b^	423.62 ^a^	13.68	<0.01	<0.01	<0.01
Crypt depth, μm	137.96	123.95	144.87	126.52	5.34	0.15	0.67	0.84
V:C	3.11 ^ab^	3.50 ^a^	2.62 ^b^	3.45 ^a^	0.12	0.01	0.22	0.33
Ileum								
Villus height, μm	318.70 ^ab^	340.07 ^a^	265.98 ^b^	344.75 ^a^	10.87	0.01	0.21	0.14
Crypt depth, μm	137.96	123.95	144.87	126.52	5.34	0.15	0.67	0.84
V:C	2.42 ^ab^	3.05 ^a^	2.23 ^b^	2.79 ^ab^	0.14	0.04	0.42	0.9

V:C, Villus height: Crypt depth. Mean and total SEM are listed in separate columns (*n* = 8). a, b mean values within a row with unlike superscript letters were significantly different (*p* < 0.05). CON pigs were fed with a basal diet; BGL pigs were fed with a BGL containing diet (500 mg/kg); ECON pigs were fed with a basal diet and challenged by ETEC; EBGL pigs were fed with a BGL containing diet (500 mg/kg) and challenged by ETEC.

**Table 4 antibiotics-11-00519-t004:** Effect of BGL supplementation on mucosal enzyme activity in weaned pigs upon ETEC challenge.

ITEM	Treatments				SEM	*p*-Value		
	CON	BGL	ECON	EBGL		BGL	ETEC	Interaction
Duodenum								
Lactase (U/L)	78.59 ^a^	66.08 ^c^	72.92 ^b^	66.03 ^c^	1.42	0.001	0.006	0.001
Sucrase (U/L)	255.48 ^a^	254.47 ^a^	234.90 ^b^	264.43 ^a^	2.76	0.001	0.16	<0.01
Maltase (U/L)	175.56 ^c^	188.19 ^b^	166.91 ^d^	197.63 ^a^	2.82	0.03	0.88	<0.01
Jejunum								
Lactase (U/L)	107.58 ^a^	103.48 ^a^	93.15 ^b^	102.80 ^a^	1.77	0.001	0.13	0.14
Sucrase (U/L)	353.32 ^a^	352.25 ^a^	328.36 ^b^	368.95 ^a^	4.43	0.01	0.56	0.07
Maltase (U/L)	286.45 ^b^	290.57 ^ab^	274.63 ^c^	298.55 ^a^	2.36	<0.01	0.57	0.007
Ileum								
Lactase (U/L)	39.85 ^a^	40.21 ^a^	31.15 ^b^	32.09 ^b^	1.22	0.72	<0.01	0.87
Sucrase (U/L)	294.08 ^bc^	315.82 ^a^	283.07 ^c^	306.39 ^ab^	3.27	<0.01	0.03	0.86
Maltase (U/L)	214.18 ^c^	229.17 ^b^	202.78 ^c^	243.26 ^a^	3.77	<0.01	0.76	0.008

Mean and total SEM are listed in separate columns (*n* = 8). a, b, c, d mean values within a row with unlike superscript letters were significantly different (*p* < 0.05). CON pigs were fed with a basal diet; BGL pigs were fed with a BGL containing diet (500 mg/kg); ECON pigs were fed with a basal diet and challenged by ETEC; EBGL pigs were fed with a BGL containing diet (500 mg/kg) and challenged by ETEC.

**Table 5 antibiotics-11-00519-t005:** Effect of BGL supplementation on intestinal microbiota and microbial metabolites in weaned pigs upon ETEC challenge.

ITEM	Treatments				SEM	*p*-Value		
	CON	BGL	ECON	EBGL		BGL	ETEC	Interaction
microbial populations (lg(copies/g))								
Total bacteria	11.18	11.03	11.18	11.21	0.05	0.5	0.35	0.34
*Escherichia coli*	8.35 ^b^	8.14 ^b^	9.88 ^a^	9.50 ^a^	0.22	0.42	0.001	0.81
*Lactobacillus*	8.28 ^ab^	8.75 ^a^	7.93 ^b^	8.60 ^a^	0.1	0.004	0.17	0.58
*Bifidobacterium*	6.09	6.28	6.2	6.09	0.13	0.88	0.88	0.59
*Bacillus*	9.16	9.18	9.09	9.03	0.04	0.8	0.18	0.6
VFA (g/g)								
Acetic acid	3.36 ^ab^	3.65 ^a^	2.59 ^b^	3.14 ^ab^	0.19	0.25	0.09	0.72
Propanoic acid	1.73 ^ab^	1.74 ^ab^	1.40 ^b^	2.00 ^a^	0.09	0.07	0.83	0.08
Butyric acid	0.78	0.95	0.82	0.78	0.05	0.59	0.6	0.41

VFA, volatile fatty acids. Mean and total SEM are listed in separate columns (*n* = 8). a, b mean values within a row with unlike superscript letters were significantly different (*p* < 0.05). CON pigs were fed with a basal diet; BGL pigs were fed with a BGL containing diet (500 mg/kg); ECON pigs were fed with a basal diet and challenged by ETEC; EBGL pigs were fed with a BGL containing diet (500 mg/kg) and challenged by ETEC.

**Table 6 antibiotics-11-00519-t006:** Experiment basal diet composition and nutrient level.

Ingredients	%	Nutrient Level	Contents
Corn	28.31	Digestible energy (calculated, MJ/kg)	14.78
Extruded corn	24.87	Crude Protein (%)	19.68
Soybean meal	8.5	Calcium (%)	0.81
Extruded full-fat soybean	10.3	Available phosphorus (%)	0.55
Fish meal	4.2	Lysine	1.35
Whey powder	7	Methionine	0.42
Soybean protein concentrate	8	Methionine + cysteine	0.6
Soybean oil	2	Threonine	0.79
Sucrose	4	Tryptophan	0.22
Limestone	0.9		
Dicalcium phosphate	0.5		
NaCl	0.3		
L-Lysine HCl (78%)	0.47		
DL-Methionine	0.15		
L-Threonine (98.5%)	0.13		
Tryptophan (98%)	0.03		
Chloride choline	0.1		
Vitamin premix ^1^	0.04		
Mineral premix ^2^	0.2		
Total	100		

^1^ The vitamin premix provided the following per kg of diet: 9000 IU of VA, 3000 IU of VD 3, 20 IU of VE, 3 mg of VK 3, 1.5 mg of VB1, 4 mg of VB 2, 3 mg of VB6, 0.02 mg of VB12, 30 mg of niacin, 15 mg of pantothenic acid, 0.75 mg of folic acid, and 0.1 mg of biotin. ^2^ The mineral premix provided the following per kg of diet: 100 mg Fe, 6 mg Cu, 100 mg Zn, 4 mg Mn, 0.30 mg I, 0.3 mg Se. The diet was formulated based on the recommendation of NRC2012.

## Data Availability

Not applicable.

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
