# Peer review of "Effect of β-Glucan Supplementation on Growth Performance and Intestinal Epithelium Functions in Weaned Pigs Challenged by Enterotoxigenic Escherichia coli"

_antibiotics, 2022, doi:10.3390/antibiotics11040519_

Round 1

Reviewer 1 Report

It is a complete and complex study. It is well structured and presented. The materials and methods are properly presented and contain all the necessary data. The results are concise and complete, explicitly presented. The conclusions are summarized and supported by the results. The references are well chosen, in sufficient numbers and present new studies in the field.
It can be published in its present form.

Author Response

Reviewer #1: It is a complete and complex study. It is well structured and presented. The materials and methods are properly presented and contain all the necessary data. The results are concise and complete, explicitly presented. The conclusions are summarized and supported by the results. The references are well chosen, in sufficient numbers and present new studies in the field.

It can be published in its present form.

Re: Thank you for your comments.

Reviewer 2 Report

The article provides useful information about the effect of β-glucan supplementation on growth performance and intestinal epithelium functions in weaned pigs challenged by Enterotoxigenic Escherichia Coli. Although, the study was in general appropriately designed and implemented, there are some points that should be corrected or clarified.

L13: “examine” instead of “probe”

L15: “assigned” instead of “divided”

L22: What about ileum?

L26-28: Please rephrase

L28: For mucin-2 also in ileum

L28: What about claudin-1, LAT-1, FATP-1, FATP-4 and SGLU-1 in duodenum?

L36: Please delete “the”

L38: “susceptibility”

L43: “previous” instead of “past”

L46: Please delete “on”

L49-50: “Among the feed additives applied in pigs, prebiotics are regarded as preferable, since they can…”

L55: “…pathogens do not adhere to the epithelium but to the sugar…”

L58: “known” instead of “looked”

L60: “induce a” instead of “produce”

L71: “BGL can also act…”

L75-76: “…and immune functions, data regarding the effect of BGL on the intestinal epithelium functions of the weaned pig exposed to ETEC are scarce. The aim of…”

L77: “examine” instead of “probe”

L78-79: “…will also help to the understanding of the mechanisms…”

L83: “significant” instead of “prominent”

L84: “among” instead of “between”

L91-92: Please delete superscript “c” and correct to tendency (no significant difference)

L96: Please also explain “GE”

L98: Please delete

L105-106: No reduction was observed

L121: Please delete “in”

L122: “resulting” instead of “result”

L123: “enhanced” instead of “heightened”

L144-146: Please explain Fig.2 and highlight the differences among groups

L152-153: “…improved the duodenal expression levels of claudin-1 and jejunal expression levels of occludin upon ETEC challenge…”

L167: Please delete “as well”

L182: Please delete “of”

L183: “contribute to the maintenance of” instead of “is helpful for maintaining”

L184: “known” instead of “looked”

L185: “significantly” instead of “distinctly”

L190: “…plasma cells. They play a key role in the immune response, and can specifically…”

L192: “are” instead of “were”

L197: “a high level”

L199: “considered as” instead of “deemed to be the”

L208: “…height was significantly reduced upon ETEC challenge, a finding that is in agreement with current studies…”

L209: Please delete “not only obviously”

L210: “and” instead of “also”

L213: “prevent” instead of “keep”

L222-224: “In the present study, in pigs challenged with ETEC the expression levels of the major TJ protein ZO-1 in duodenum and ileum were reduced; however…”

L227-228, 256: “A previous study…”

L237: “Moreover” instead of “Whereas”

L239-240: “…and CAT1 was significantly improved due to BGL treatment…”

L242: Please delete “especially”

L249: Please rephrase

L250: Please delete “enormously”

L258, 259: Propanoic or propionic?

L265: “The present experiment was approved…”

L271: “A total of…”

L272: “allocated” instead of “designed”

L273: “design to” instead of “arrangement of”

L276-278: “…recommended by NRC [58]. Pigs were individually housed in 1.5 × 0.7 m2 metabolism cages and had free access to feed and water. Temperature was controlled between 27-30°C and relative humidity at 65 ± 5%. The experiment…”

L280: Please explain “LB”

L303: Please remove the part of “Growth performance evaluation” before the part of “Sample collection”

L304: What do you mean by first feeding? In L277-278, you state that pigs were ad libitum fed.

L306: “calculated” instead of “counted”

L331: Please delete “and”

L332: Please rephrase

L333: “collected” instead of ‘taken”

L337: “performed according” instead of “referred”

L358: Propionic or propanoic?

L388: “variation” instead of “vary”

Author Response

Point to point response to reviewer’s comments

Thank you very much for giving us an opportunity to submit a revised version of our manuscript. We appreciate for your comments and suggestions concerning our manuscript. These comments are valuable and helpful for revising and improving our manuscript, and we revised the manuscript in accordance with the detailed comments and suggestions. All revisions are highlighted in red in the text. The point-by-point revisions to the comments and suggestions are listed as follows:

Reviewer #2: The article provides useful information about the effect of β-glucan supplementation on growth performance and intestinal epithelium functions in weaned pigs challenged by Enterotoxigenic Escherichia Coli. Although, the study was in general appropriately designed and implemented, there are some points that should be corrected or clarified.

(1)  L13: “examine” instead of “probe”

Re: Thanks for your comments. The words you mentioned have been revised in the revised manuscript.

(2)  L22: What about ileum?

Re: Thanks for your comments. We are sorry for the missing description of the ileum-related data, we have added the corresponding description in the revised manuscript.

(3) L26-28: Please rephrase

Re: Thanks for your comments. The description in this section has been rephrase.

(4) L28: What about claudin-1, LAT-1, FATP-1, FATP-4 and SGLU-1 in duodenum?

Re: Thanks for your comments. We have added the corresponding description in the revised manuscript. See Section (3) for the revised content.

(5) L49-50: “Among the feed additives applied in pigs, prebiotics are regarded as preferable, since they can…”

Re: Thanks for your comments. We are sorry that the language in this article is not refined enough, and all similar suggestions in the article have been revised according to your suggestion in the revised manuscript.

  • L96: Please also explain “GE”

Re: Thanks for your comments. “GE” is the short of “gross energe”, we have added the description in the revised manuscript.

  • L105-106: No reduction was observed

Re:Thanks for your comments. We are sorry for the mis-description of the results, which has been re-described according to your suggestion in the revised manuscript.

  • L144-146: Please explain Fig.2 and highlight the differences among groups

Re: Thanks for your comments. Explanation for Fig. 2 and highlight the differences among groups as follows: Duodenum, jejunum and ileum in the CON group revealed a normal appearance with regular intestinal villus structure; duodenum, jejunum and ileum in the BGL group, there were no obvious damages were found. However, in the ECON group, the intestinal lesions were obvious and some intestinal villi were necrotic and shed, or even disappeared, especially in jejunum; duodenum, jejunum and ileum in the EBGL group, there were no obvious intestinal lesions and only few of intestinal villi were necrotic and shed. The explanation appeared on L146-150 in the revised manuscript.

  • L249: Please rephrase

Re: Thanks for your comments. The sentence has been rephrased in the text.

  • L303: Please remove the part of “Growth performance evaluation” before the part of “Sample collection”

Re: Thanks for your comments. It has been removed in the revised manuscript.

  • L304: What do you mean by first feeding? In L277-278, you state that pigs were ad libitum fed.

Re: Thanks for your comments. In this study, the piglets were fed manually instead of an automatic feeding system. During the experiment, we fed four times a day to ensure that the diet is accessible for the piglets at any time. Animals needed to be fasted for 12 hours before blood collection, and then fed piglets after blood collection to ensure that fresh cecal chyme samples can be collected, so it is called first feeding. To avoid ambiguity, we removed the words “first feeding”.

Reviewer 3 Report

Dear Authors,

The article tackles very important topic in veterinary medicine. Increasing AMR provokes the need to look for alternatives to antimicrobials. According to the presented results, β-glucan offers numerous benefits for piglets facing the challenge of post-weaning diarrhea.

The study is well designed and written. The description dose not rise any bigger concerns. However, I would like to ask for some explanations and if existing for more results.

1/ What was the reason of weaning the piglets at such a young age? 21d it is very early.

2/ Did you score the clinical sign of animals (including diarrhea intensity and duration)?

3/ Why did you study the expression of only three TJ-related genes (ZO-1, claudin-1 and occluding) and why those three? Most importantly, did you check the expression of the proteins, too?

Author Response

Reviewer #3: The article tackles very important topic in veterinary medicine. Increasing AMR provokes the need to look for alternatives to antimicrobials. According to the presented results, β-glucan offers numerous benefits for piglets facing the challenge of post-weaning diarrhea. The study is well designed and written. The description dose not rise any bigger concerns. However, I would like to ask for some explanations and if existing for more results.

  • What was the reason of weaning the piglets at such a young age? 21d it is very early.

Re: Thanks for your comments. The main reason for weaning at 21 days is economic value of the sow. The experimental animals are commercial crossbred pigs (DLY) purchased from a large-scale pig farm. For higher economic value, pig farms make full use of the reproductive performance of the sows, they shorten the weaning age.

Weaning is one of the biggest stresses events in life of pigs. Without the protection of passive immunity and changes in nutritional sources, piglets are prone to intestinal and immune system dysfunction. Therefore, nutritional strategies are commonly used to reduce the adverse effects of weaning stress to improve the gut health, growth performance and health status of nursery pigs, which coincides with the purpose of this experiment.

  • Did you score the clinical sign of animals (including diarrhea intensity and duration)?

Re: Thanks for your comments. We scored animals for diarrhea intensity throughout the experiment, and during E. coli challenge period, we recored both diarrhea intensity and diarrhea duration. Diarrhea rate was calculated according to the formula where diarrhea rate (%) = number of pigs with diarrhea within a treatment/(number of pigs Í total observational days) Í 100, where “number of pigs with diarrhea” was the total number of pigs with diarrhea observed each day (Huang et al., 2004; Sun et al., 2008).

In this experiment, compared with the CON group, the intestinal morphological structure was damaged, the production of serum inflammatory factors (e.g. IL-1β, IL-6) increased, and intestinal permeability indicators D-lactate(data not shown) increased in the ETEC challenge pigs. In addition, the abundance of E. coli in the cecal digesta also significantly increased. We believed that the Escherichia coli (ETEC) challenge model in weaned pigs well-established, so indicators related to diarrhea do not need to be shown in addition.

  • Why did you study the expression of only three TJ-related genes (ZO-1, claudin-1 and occluding) and why those three? Most importantly, did you check the expression of the proteins, too?

Re: Thanks for your comments. Although there are many TJ proteins, these three  (ZO-1, claudin-1 and occludin) are reported the most common and their importance to barrier function is widely demonstrated (Buckley et al,2019; Mu et al,2013;Shen et al,2011). We are sorry that we did not check the expression of the proteins as it is difficult to obtain all these commercial antibodies for pigs. However, we used real-time quantitative PCR (qPCR) to quantify gene expression. Real-time  quantitative PCR technology is widely used in the quantification of gene expression levels (Yu et al,2020; Luo et al,2021). Real-time quantitative PCR detection technology is a technology that integrates PCR technology, fluorescent labeling technology, laser technology, and digital imaging technology. The advantage of this method is the high detection sensitivity. A fluorescent signal is generated corresponding to each amplification product, which can be directly quantified by detecting the fluorescent signal. And the versatility of fluorescence quantification is good, suitable for most types of quantitative reactions, and melting curve analysis can be performed to test the specificity of amplification reactions. In addition, amplification and detection can be detected in the same tube, which is not easy to contaminate.